# Evaluation of cortical thickness and cortical thickness index in the proximal femur—CT, dual energy absorptiometry (DXA), trabecular bone score (TBS) –

Xiao Ma[1], Ik Yang[2], Sewon Lee[3], Jungyoun Kim[4], Hyunjin Park[4], Younghyun Yoon[4], Jihyo Hwang[4]*

1 Department of Orthopedics, China-Japan Union Hospital of Jilin University, Changchun, Jilin, PR China, 2 Department of Radiology, Gangnam Sacred Heart Hospital, Hallym University School of Medicine, Seoul, Republic of Korea, 3 Department of Orthopedics, College of Medicine, Yeouido St. Mary's Hospital, Catholic University School of Medicine, Seoul, Republic of Korea, 4 Department of Orthopaedics, Gangnam Sacred Heart Hospital, Hallym University School of Medicine, Seoul, Republic of Korea

* hwangjihyo36@gmail.com

**Data Availability Statement:** We uploaded our patient data on figshare and it's DOI is 10.6084/m9.figshare.24755277.

## Abstract

### Purpose

The purpose of this study was to assess the anatomical size of proximal femur in South Korea. This study measured cortical thickness and cortical thickness index (CTI) based on computed tomography (CT) and additionally, evaluated the T-score and trabecular bone score (TBS) based on the dual energy X-ray absorptiometry (DXA).

### Materials and methods

This retrospective study is a cross-sectional study based on data from 600 patients aged from 20 to 93 years during the time from 2011 to 2021 were enrolled and selected the patients who did the examination of both pelvic CT and DXA scan. Age, sex, BMI, T-score, TBS, cortical thickness, CTI and the size of proximal femur were analyzed. Among these patients, 200 patients each corresponding to femoral neck fracture group(N = 200), trochanteric fracture group(N = 200), and non-fracture group(N = 200) were randomly selected and studied. The differences of three groups were compared statistically.

### Results

Mean outer diameter of proximal femur was 24.34 mm, inner diameter of proximal femur was 15.28 mm, cortical thickness was 4.55 mm and CTI was 0.37 at the lesser trochanter (LT) level. The outer diameter was 24.00 mm, inner diameter of proximal femur was 13.04 mm, cortical thickness was 4.97mm and CTI was 0.44 at 3cm below LT. In the hip fracture group, T-score of hip, outer diameter of proximal femur and cortical thickness at LT were lower than non-fracture group. BMI, T-score of spine, T-score of hip, inner diameter at 3cm below LT, CTI of LT and TBS were lower in femoral neck fracture group compared to the trochanteric fracture group.

**Funding:** The authors received no specific funding for this work.

**Competing interests:** The authors have declared that no competing interests exist.

## Conclusion

Analysis of 600 patients of pelvic CT might be a representative of real size of proximal femur in South Korea. Outer diameter of proximal femur at LT and cortical thickness at LT level were significantly lower in hip fracture group. Narrower outer diameter of proximal femur and thinner cortical thickness at LT level from the CT might be a risk factor of hip fracture.

## Introduction

Currently, the most affected area in the field of orthopedic surgery is osteoporosis because the incidence of traumatic fractures is declining and that of osteoporotic fractures is increasing [1]. A senile fracture is associated with osteoporosis, with an increasing trend in accordance with aging [2]. Moreover, hip fracture in the elderly represents osteoporotic fracture. A study reported a clear relationship between decreased bone mineral density (BMD) and an increased risk of hip fracture [3]. Regarding prevention of osteoporotic fractures, the Fracture Risk Assessment Tool (FRAX) can reveal trends related to osteoporosis worldwide [4]. FRAX scores are measured by dual-energy X-ray absorptiometry (DXA) T-score or Trabecular Bone Score (TBS). The TBS, in particular, can more accurately represent the microstructure of the spine, providing a more precise representation of the skeletal microstructure than the T-score [5].

In addition, a recent study suggests that proximal femoral morphology plays an important role in hip fracture etiology [6]. The Dorr classification, neck–shaft angle, neck length, cortical thickness, cortical thickness index (CTI), anatomical bowing, which are sort of morphology and geometry of the femur [7].

Proximal femoral geometry can be measured using digital photographs obtained from cadavers, radiographs, or computed tomography scans [8–10]. The study of the structure of the proximal femur and of structural changes with aging are also among the studies aimed at preventing hip fractures [11]. In particular, Changes in cortical thickness and CTI have been studied on various bones, and several studies have reported on the relationship between these data and fractures [7,12–15]. In 2007, Sah et al. proposed CTI as an easy-to-assess and valid tool to assess the bone density of the proximal femur based on plain hip radiographs [16]. This study assessed cortical thickness and CTI based on CT scans.

The purpose of this study was to assess the proper size of the proximal femur using CT in South Korea. The more accurate size of the bone can be currently assessed using CT, and it may be significant compared with plain radiography used in many studies, which could be magnified. This study was meaningful because Korean data so far have been studied on the bowing of the whole femur, but there are no data on the proximal femur. The number of patients considered (N = 600) represents a large population so far, so these data can be representative for the evaluation of the geometry of the proximal femur in an East Asian population. These data can also be correlated to the fracture risk and basic data of three-dimensional (3D) printing of implants in the future.

## Materials and methods

This retrospective study is a cross-sectional study based on data from 600 patients aged 20–93 years in South Korea who visited Gangnam Sacred Heart Hospital.

All patients were enrolled during 2011–2021. The data of patients who underwent both pelvic CT (Somatom Definition Flash, Siemens Healthcare, Germany) and DXA (Hologic, Inc,

Budford, MA, USA) were collected. CT image was recorded using a setting of 2 mm thickness. For patients with hip fractures, CT images of their healthy limbs were evaluated.

However, among the patients who visited our hospital without fractures, it was very difficult to find patients who performed all the necessary tests when we studied. Therefore, among patients without fractures, 200 patients who had completed the examination were collected and compared with patients with fractures.

The study was approved by the Institutional Review Board of Gangnam Sacred Heart Hospital, Hallym University (IRB No. 2022-05-006), where the investigation was performed. All methods were carried out in accordance with relevant guidelines and regulations. The consent for patient data was waived by the ethics committee.

### Patient population

This study was conducted on patients whose demographic data, CT, and DXA were all collected among patients who visited the hospital during the above period. Among these patients, 200 patients each corresponding to femoral neck fracture group(N = 200), trochanteric fracture group(N = 200), and non-fracture group(N = 200) were randomly selected and studied.

The exclusion criteria were a history of congenital hip-related pathologies and lack of any information needed for analysis. Data require for analysis include age, sex, and body mass index (BMI)

### Data collection

Demographic data such as age, sex, and body mass index (BMI) were analyzed. Using the DXA system, T-score was obtained for the proximal femur and lumbar spine (Fig 1). The data of each of the three groups of patient were recorded, and a comparative analysis was performed: femoral neck fracture(Fig 2), trochanteric fracture(Fig 3), and non-fracture groups (N = 200 each). The selected 600 patients were also divided into two groups: hip fracture group (N = 400) and non-fracture group (N = 200). Propensity score matching was performed to compare the data of the two groups.

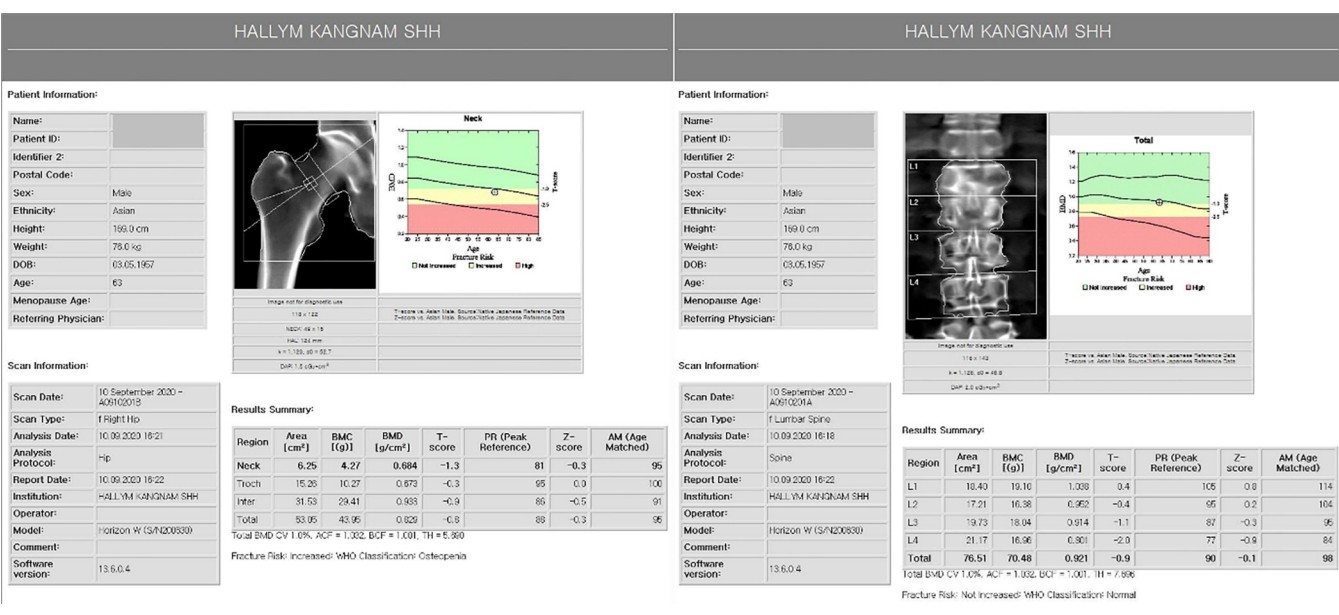

**Fig 1.**

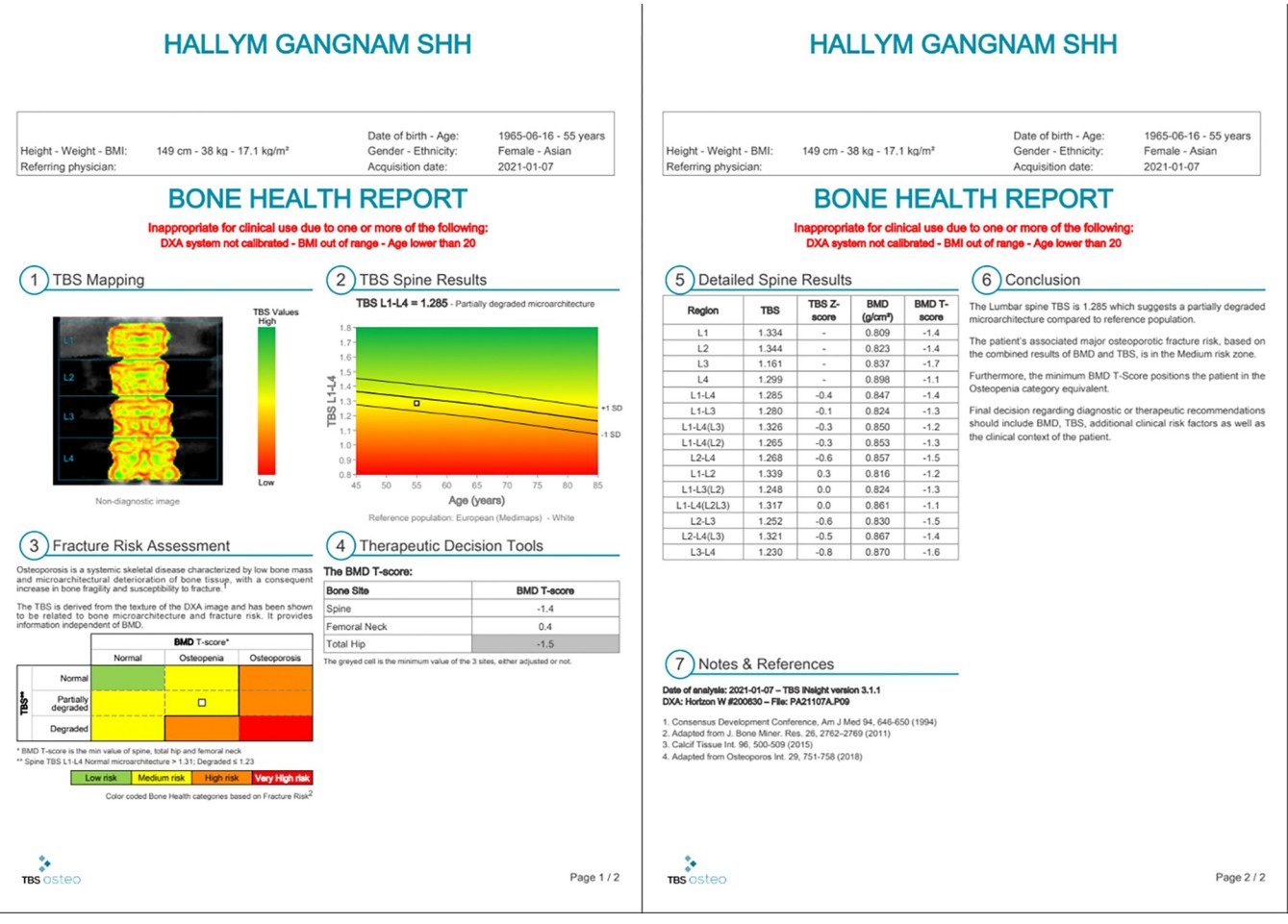

**Fig 2.**

TBS (TBS Osteo software 3.1) was also calculated using DXA data (Fig 4). TBS is a textural index that evaluates pixel gray-level variations on the lumbar spine DXA image, providing an indirect index of trabecular microarchitecture. In this study, two groups were formed: trochanteric fracture and femoral neck fracture groups. The TBS of L1–L4, TBS of L1, TBS of L2, TBS of L3, TBS of L4, Z-score of TBS, minimum T-score, and maximum T-score were analyzed and compared.

Regarding cortical thickness, the lesser trochanter (LT) level, 3 cm below the LT level, and CTI were selected because many pelvic CT scans did not contain sufficient length from the LT level, with 3–8 cm of length from the LT level being available on pelvic CT. Data from two levels were also compared. All measurements were performed by two orthopedic surgeons (X. Ma, J.Hwang). Interobserver and interobserver reliabilities were calculated using Person's coefficients (r). The final data were calculated by using the average of the data from the two surgeons.

Using the Picture Archiving and Communication System (M6, INFINITT Healthcare, South Korea), the computer-based measurements were performed. For the appropriate and consistent collection of data, we defined the center of the proximal femoral shaft as follows. First, the largest diameter of the femoral head was obtained (Fig 5). Next, the neck–shaft angle was calculated and the perpendicular line to the vertical axis of the femur shaft was evaluated.

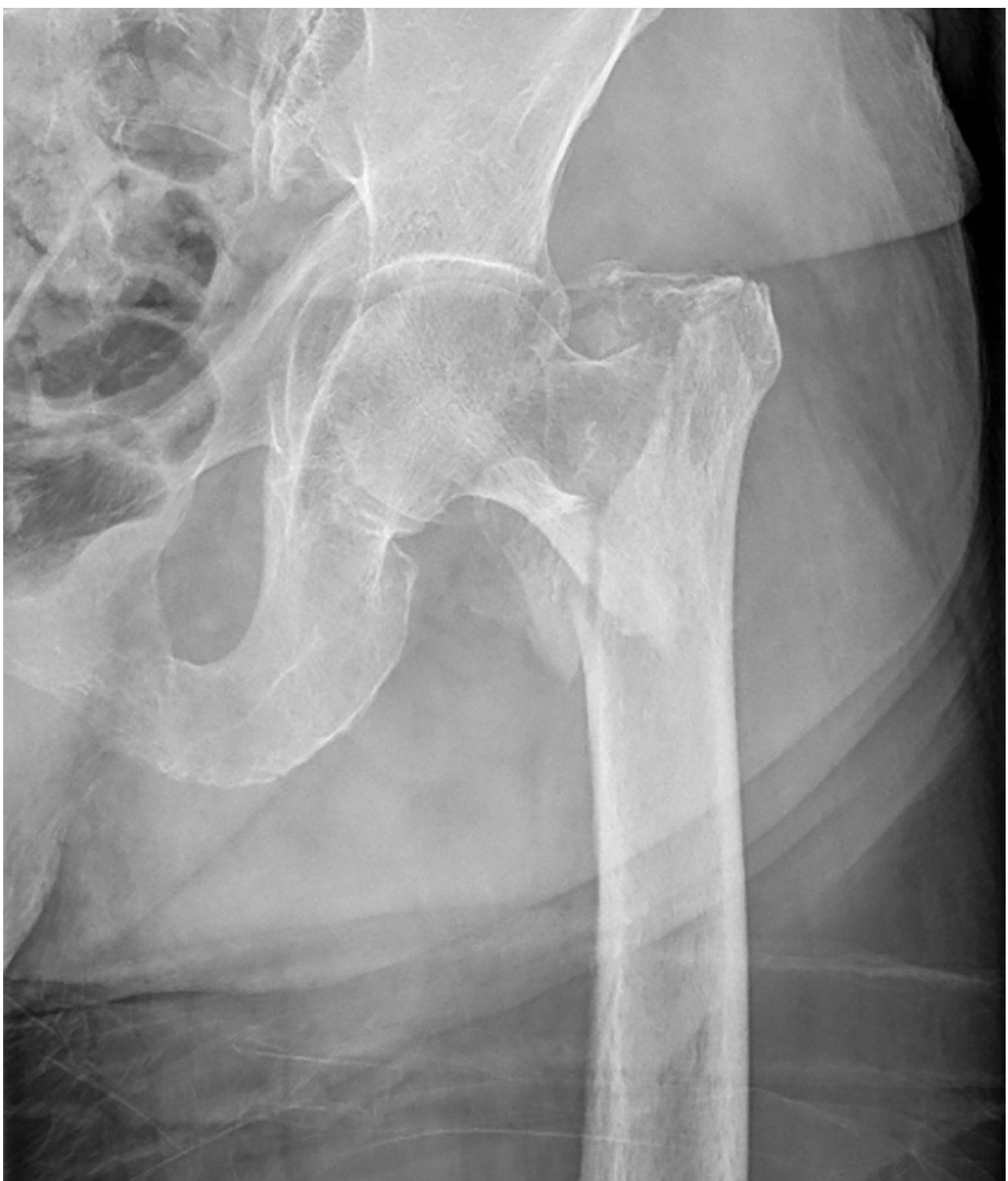

**Fig 3.**

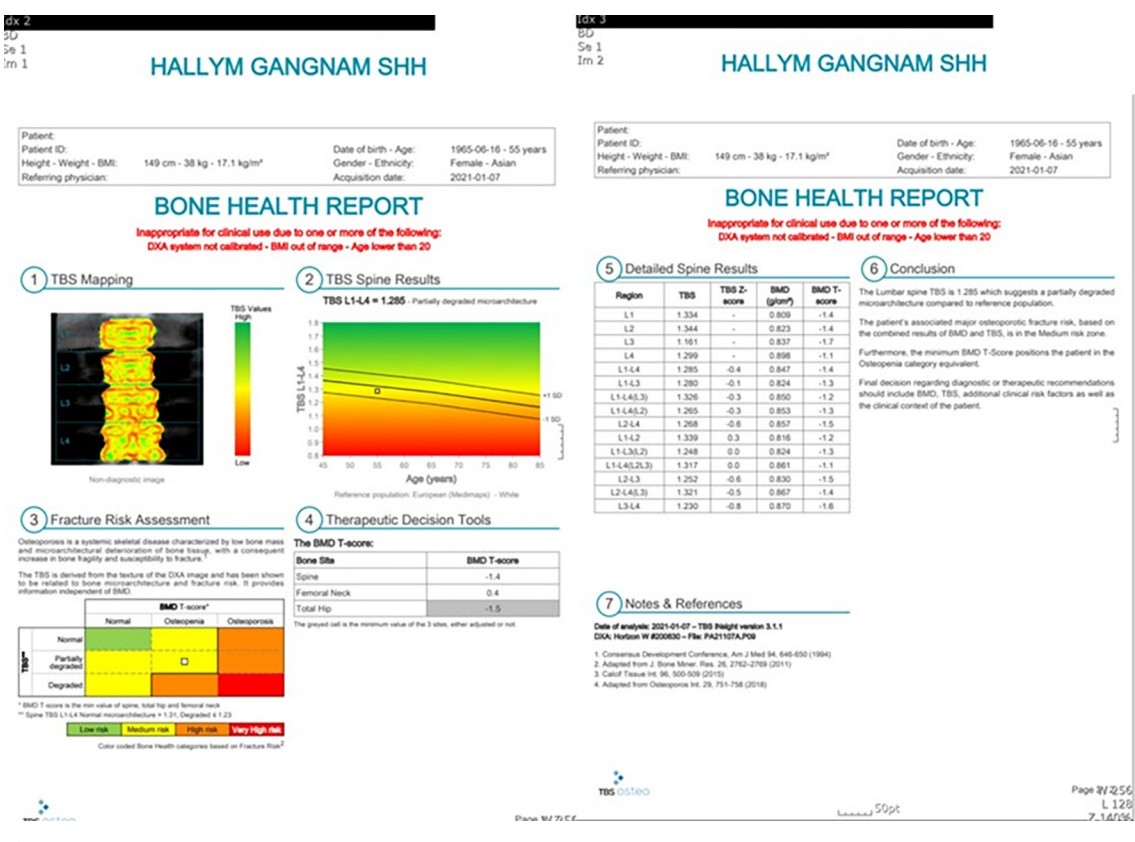

**Fig 4.**

For example, diameters of 36.72, 37.67, 38.16, 37.67, and 35.86 mm were obtained, and the largest size of the femoral head, 38.16 mm, was selected as the standard that represents the center of the femoral head in the coronal view on CT for appropriate and consistent definition of the center of the proximal femur (Fig 6).

Then, the inner diameter and outer diameter of the proximal femur were measured as 18.24 and 26.17 mm, respectively, at the LT level (Fig 7). Cortical thickness at the LT level and 3 cm below LT level, and CTI at LT level and 3 cm below LT level were analyzed. Inner diameter of 14.27 mm of and outer diameter of 24.58 mm at the level of 3 cm below LT were observed (Fig 8). CTI was defined as the ratio of cortical width (outer diameter) minus endosteal width (inner diameter). In previous studies, these parameters were originally measured at a level of 10 cm below the midpoint of the LT [7,9]. The current study selected the LT and 3 cm below LT because 10 cm below the LT was not available on pelvic CT. On femoral radiographs, femoral cortical thickness can be measured in three regions: 5 cm, 12.5 cm below the LT, and in the region of maximal cortical thickness [17]. In this study, the LT level and 3 cm below the LT were selected to calculate CTI. A CTI at two different levels was calculated. The LT level was considered the first level [CTI = (A − B / A)] and the second level was 3 cm below the LT level [CTI = (C − D / C)]. A and C represent outer diameters, whereas B and D represent inner diameters (Fig 9).

## Statistical analysis

Date analysis was performed using IBM SPSS Statistics for Windows, version 27 (IBM Corp., Armonk, NY, USA). All measurement data were described by a combination of

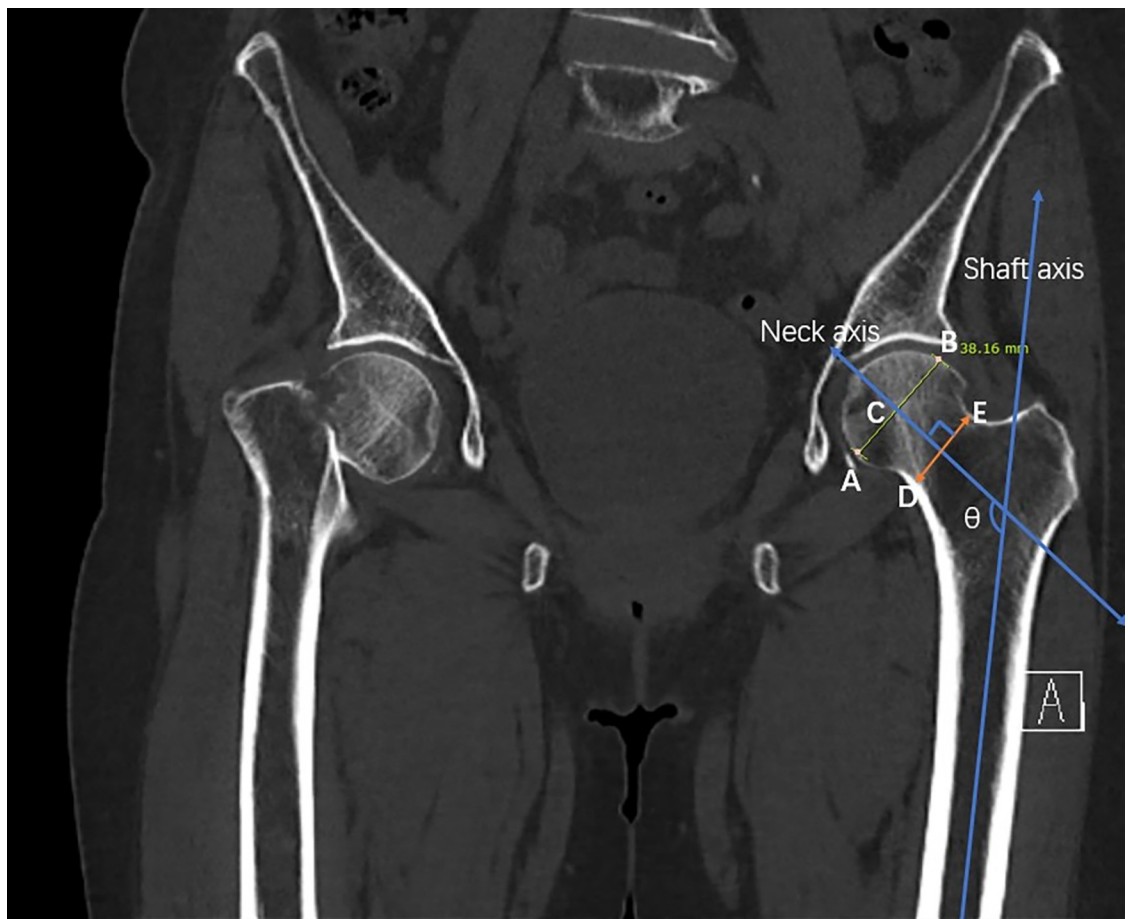

**Fig 5.**

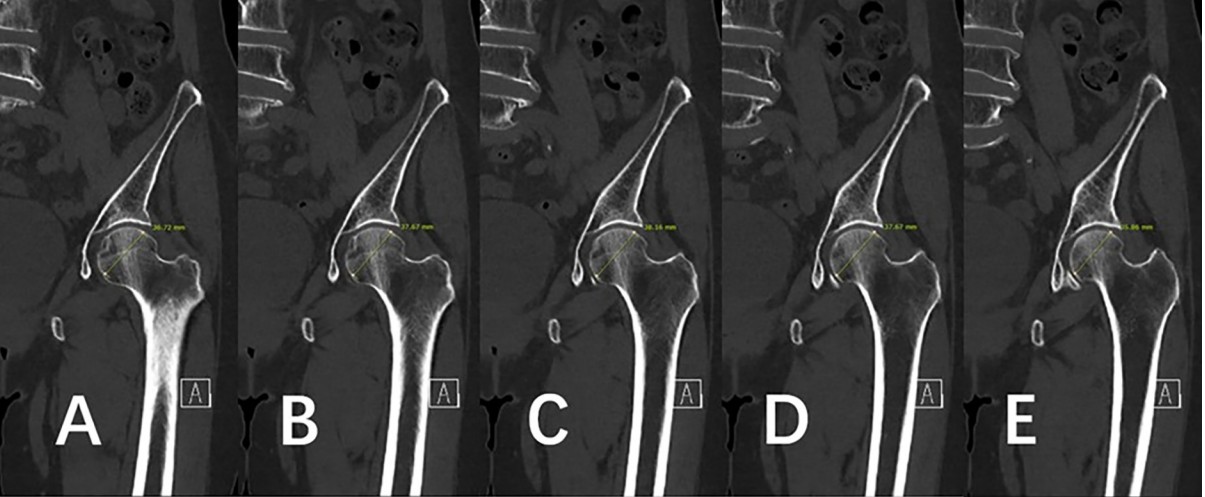

**Fig 6.**

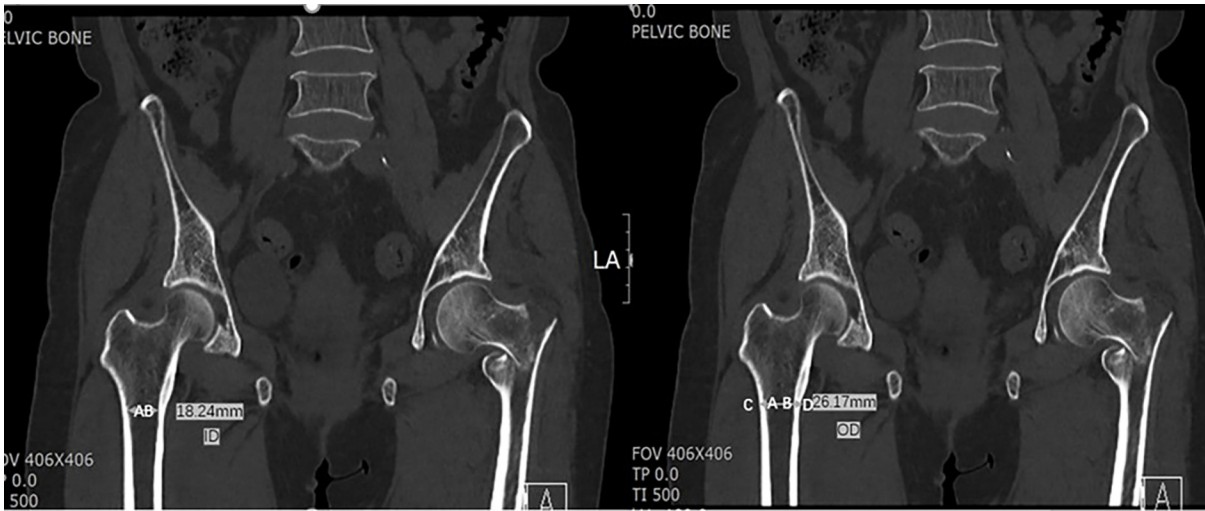

**Fig 7.**

mean ± standard deviation, range, and 95%CI. The OD, ID, CTI, TBS parameters between fracture group and non fracture group or femoral neck fracuture and trochanteric fracture group were compared by Student's t-test but the parameter between 6 age groups for Tables 5, 6 was compared by ANOVA.

P-values of <0.05 were considered statistically significant.

## Results

### 1. Average data from CT measurements in the proximal femur

The average data obtained after completing the measurement of data of all 600 patients are as follows: The outer diameter of the cortex at the LT level was 24.34 mm (range, 6.24–40.5) and at 3 cm below it was 24.00 mm (range, 14.05–34.77). The inner diameter of the proximal femur at the LT level was 15.28 mm (range, 8.18–27.75) and at 3 cm below it was 13.04 mm

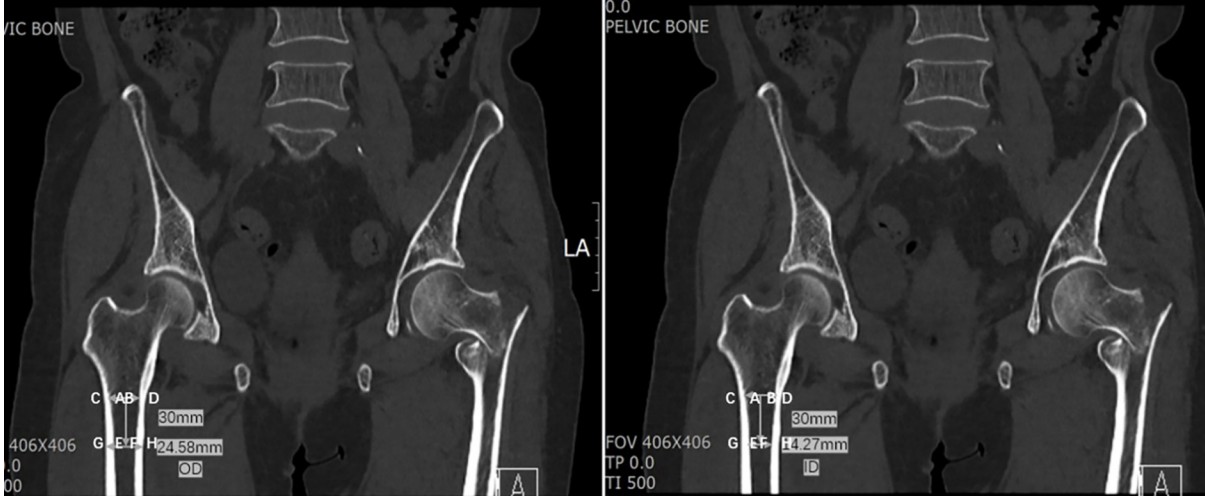

**Fig 8.**

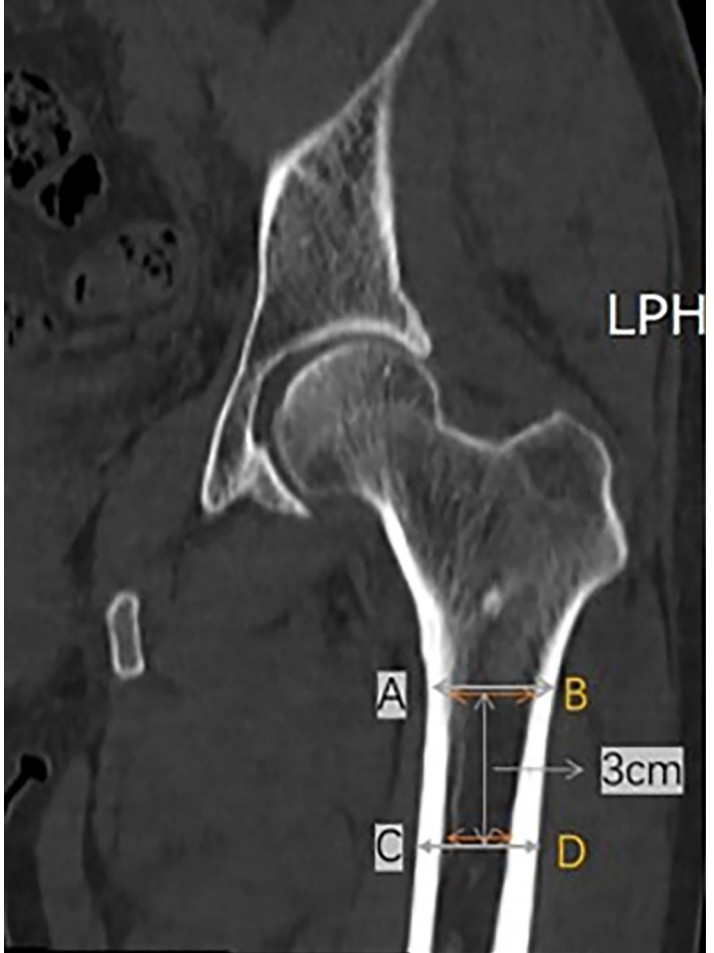

**Fig 9.**

(range, 7.45–25.14). Cortical thickness at the LT level was 4.55 mm (range, 2.02–8.45) and at 3 cm below it was 4.97 mm (range, 0.23–10.45). The CTI at the LT level was 0.37 mm (range, 0.14–0.61) and at 3 cm below it was 0.44 mm (range, 0.17–4.03) (Table 1).

The outer and inner diameters were both thicker at the LT level than at 3 cm below it (p < 0.001). Cortical thickness and CTI at 3 cm below the LT level were thicker than at the LT level (p < 0.001).

## 2. Age- and sex-adjusted comparison between the non-fracture and hip fracture groups

The outer diameter and cortical thickness of the LT level were significantly different from the DXA T-score of the hip. This suggests that a lower T-score of the hip and narrower outer diameter and lower cortical thickness of the LT could be risk factors for hip fracture (Table 2).

## 3. Comparison between the trochanteric fracture and femoral neck fracture groups

A significant difference was observed in the ranges of BMI, T-scores of the spine and hip, inner diameter at 3 cm below the LT level, and CTI of the LT level. It suggests that higher BMI,

**Table 1. Mean values of parameters measured through CT(n = 600).**

| | |
|---|---|
| **OD of the proximal femur at the LT level** | **24.34 mm (16.24 ~ 40.5)** |
| **OD of the cortex at 3 cm below the LT level** | 24.00 mm (14.05 ~ 34.77) |
| **ID of the proximal femur at the LT level** | 15.28 mm (8.18 ~ 27.75) |
| **ID of the proximal femur at 3 cm below the LT level** | 13.04 mm (7.45 ~ 25.14) |
| **Cortical thickness at the LT level** | 4.55 mm (2.02 ~ 8.45) |
| **Cortical thickness at 3 cm below the LT level** | 4.97mm (0.23 ~ 10.45) |
| **CTI at the LT level** | 0.37 (0.14 ~ 0.61) |
| **CTI at 3 cm below the LT level** | 0.44 (0.17 ~ 4.03) |

The outer and inner diameters were both thicker at the LT level than at 3 cm below it (p < 0.001). Cortical thickness and CTI at 3 cm below the LT level were thicker than at the LT level (p < 0.001).

Continuous data are shown as average data(min~max).

OD: Outer diameter, ID: Inner diameter, LT: Lesser trochanter, CTI: Cortical thickness index, CT: Computed tomography.

lower T-scores of the spine and hip, thicker inner diameter at 3 cm below the LT level, and lower CTI were more better predictors of trochanteric fracture than femoral neck fracture (Table 3).

## 4. Comparison of TBS in the trochanteric fracture and femoral neck fracture groups

Comparison of the trochanteric fracture and the femoral neck fracture groups revealed that lower TBS of L1–L4, T-score of TBS, TBS of L1, TBS of L2, TBS of L3, and TBS of L4 could be risk factors for femoral neck fractures (Table 4).

**Table 2. Comparison of the parameter between non-fracture(n = 89) and fracture groups(n = 89).**

| | Non-fracture (N = 89) | Hip fracture (N = 89) | P-value |
|---|---|---|---|
| **Age, years** | 70.92(±9.00) (60~91) | (71.15±6.69) (60~87) | 0.843 |
| **Sex** | Female | Female | |
| **BMI, kg/m²** | 23.67(±3.66) | 22.43(±2.70) | 0.301 |
| **T-score of the spine** | -2.24(±1.16) (-5.2~0.5) | -2.23(±1.27) (-5~1) | 0.63 |
| **T-score of the hip** | -2.31(±1.08) (-4.7~1) | -2.62(±0.823) (-5.5~-0.8) | **0.039** |
| **OD of the proximal femur at the LT level** | 24.73 mm (±2.69) (17.90~34.03) | 23.71mm (±2.35) (16.77~29.29) | **0.008** |
| **OD of the proximal femur at 3 cm below the LT level** | 22.56mm (±2.71) (17.89~28.46) | 22.67mm (±2.19) (14.05~30.16) | 0.754 |
| **ID of the proximal femur at the LT level** | 15.13mm (±2.59) (10.23~22.13) | 14.82mm (±2.81) (10.09~20.16) | 0.451 |
| **ID of the proximal femur at 3 cm below the LT level** | 12.50mm (±2.29) (7.73~18.12) | 13.08mm (±1.92) (9.22~18.36) | 0.07 |
| **Cortical thickness at the LT level** | 4.80mm (±1.05) (2.26~8.42) | 4.35mm (±1.02) (2.26~7.09) | **0.005** |
| **Cortical thickness at 3 cm below the LT level** | 4.97mm (±1.30) (2.35~8.71) | 4.79mm (±1.10) (2.53~7.28) | 0.256 |
| **CTI at the LT level** | 0.38(±0.76) (0.21~0.61) | 0.36(±0.77) (0.19~0.58) | 0.065 |
| **CTI at 3 cm below the LT level** | 0.44(±0. 08) (0.25~0.65) | 0.42(±0.09) (0.22~0.6) | 0.346 |

The outer diameter and cortical thickness of the LT level in non fracture group were significantly greater than hip fracture group(P < 0.05).

OD: Outer diameter, ID: Inner diameter, LT: Lesser trochanter, CTI: Cortical thickness index.

**Table 3. Comparison of the parameter between trochanteric fracture(n = 200) and femoral neck fracture groups(n = 200).**

| | Trochanteric fracture (N = 200) | Femoral neck fracture (N = 200) | P value |
|---|---|---|---|
| Age | 76.11(±10.72) (32~96) | 74.03(±11.86) (21~96) | 0.069 |
| sex | Male: Female 67:132 (33.7%:66.3%) | Male: Female 73:120 (37.6%:62.8%) | 0.414 |
| BMI | 22.99 (±4.35) | 21.17 (±2.86) | **0.042** |
| T-score of spine | -2.39 (±1.38) | -2.07 (±1.44) | **0.028** |
| T-score of hip | -2.8 (±1.01) | -2.56 (±0.92) | **0.018** |
| OD of proximal femur at LT | 24.38 (±2.9) | 24.39 (±3.08) | 0.997 |
| OD of proximal femur at 3cm below LT | 23.64 (±3.04) | 23.06 (±2.73) | 0.05 |
| ID of proximal femur at LT | 15.86 (±2.53) | 15.31 (±3.03) | 0.054 |
| ID of proximal femur at 3cm below LT | 15.86 (±2.53) | 13.23 (±2.37) | **0.001** |
| Cortical thickness at LT | 4.29 (±1.11) | 4.49 (±1.11) | 0.081 |
| Cortical thickness at 3cm below LT | 4.83 (±1.46) | 4.91 (±1.26) | 0.54 |
| CTI at LT | 0.351 (±0.768) | 0.369 (±0.777) | **0.018** |
| CTI at 3cm below LT | 0.416 (±0.204) | 0.424 (±0.858) | 0.621 |

The BMI, T-scores of the spine and hip, inner diameter at 3 cm below the LT level, and CTI of the LT level in femroal neck fracture group were significantly greater than trochanteric fracture group(P<0.05), especially ID of proximal femur at 3cm below LT(P<0.01).

OD: Outer diameter, ID: Inner diameter, LT: Lesser trochanter, CTI: Cortical thickness index.

## 5. Average data according to age

All patients in the study were grouped into six groups based on age, ranging from below 50s, 60s, 70s, 80s, 90s, and above 90s. The data from DXA, including BMI and T-scores of the spine and hip, are shown in Table 5. The average outer diameter of the proximal femur at the LT level and 3 cm below it, the inner diameter of the proximal femur at the LT level and 3 cm below it, the cortical thickness at the LT level and 3 cm below it, and CTI at the LT level and 3 cm below it were analyzed according to the different age groups (Table 6). The whole data set, except BMI, was statistically significant.

## Discussion

To the best of our knowledge, the current study is the first to show the data of 600 patients and evaluate the proximal femoral geometry based on CT findings. Six hundred patients might be

**Table 4. Comparison of TBS in L1-L4 in the trochanteric fracture and femoral neck fracture groups.**

| | Trochanteric fracture (N = 160) | Femoral neck fracture (N = 90) | P-value |
|---|---|---|---|
| TBS of L1–L4 | 1.2502 (±0.08) | 1.2133 (±0.10) | **0.006** |
| T-score of TBS | 0.1868 (±0.85) | 0.1630 (±1.13) | **0.021** |
| TBS of L1 | 1.2070 (±0.11) | 1.1726 (±0.17) | **0.003** |
| TBS of L2 | 1.2650 (±0.11) | 1.2248 (±0.14) | **0.023** |
| TBS of L3 | 1.2580 (±0.10) | 1.2247 (±0.11) | **0.026** |
| TBS of L4 | 1.2629 (±0.10) | 1.2283 (±0.11) | **0.023** |
| Lowest TBS | 1.1585 (±0.11) | 1.1083 (±0.14) | **0.005** |
| Highest TBS | 1.3347 (±0.08) | 1.3087 (±0.11) | **0.004** |

All parameter was signicantly lower in femoral neck fracture group(P < 0.05).

P <0.05 is considered statistically significant.

TBS: Trabecular bone score.

**Table 5. Comparison of average BMI and T-scores of the spine and hip between 6 groups based on age.**

| Age, years | BMI, kg/m² | T-score of the spine | T-score of the hip | | | | |
|---|---|---|---|---|---|---|---|
| <50s | 21.97(±1.49) | -1.158(±1.18) | -0.965(±0.89) | | | | |
| 51~60s | 20.32(±4.24) | -1.652(±1.15) | -1.593(±0.77) | | | | |
| 61~70s | 23.65(±6.59) | -1.807(±1.27) | -2.038(±0.93) | | | | |
| 71~80s | 22.79(±3.25) | -2.307(±1.32) | -2.567(±0.79) | | | | |
| 81~90s | 22.03(±3.91) | -2.409(±1.34) | -3.033(±0.96) | | | | |
| >90s | 21.81(±3.03) | -3.147(±1.81) | -3752(±0.78) | | | | |
| N = 600 | 22.38 | -2.19 | -2.36 | | | | |
| P-value | 0.456 | **0.000** | **0.000** | | | | |

$P < 0.05$ is considered statistically significant.
BMI: Body mass index.

sufficient to evaluate the exact geometry of the proximal femur. CT-based analysis is one of the most accurate methods for normal anatomy determination [18]. In addition to CT base, the comparison and analysis of TBS and T-score from DXA in hip fracture patients are unique properties in this study. This study also discovered age-related bony change, which may predict the hip fracture risk and could be a useful data point in the 3D printing of implants for hip fracture patients. Cortical thickness and CTI were evaluated on plain radiography in previous studies [7,13]. CT is more accurate in terms of the geometry of the proximal femur. Many studies are designed to measure and calculate the cortical thickness and CTI at 10 cm below LT on plain radiography [19–23]. This study not only measured cortical thickness and CTI on CT scan but also developed and designed a new experimental method that measures the data at the LT level and 3 cm below it, which is the first and unique among other studies. For obtaining more data from CT, the researchers selected the level of LT and 3 cm below it because not all CT procedures include the far distal part of the proximal femur. There are

**Table 6. Comparison of average OD, ID, cortical thickness and CTI of the LT and 3cm below LT between 6 groups based on age.**

| Age | OD of the proximal femur at the LT level | OD at 3 cm below the LT level | ID of the proximal femur at the LT level | ID of the proximal femur at 3 cm below the LT level | Cortical thickness of LT | Cortical thickness 3 cm below the LT level | CTI of LT | CTI 3 cm below the LT level |
|---|---|---|---|---|---|---|---|---|
| <50s | 24.63(±2.25) | 22.55(±2.70) | 14.12(±2.69) | 11.91(±2.66) | 5.25(±0.92) | 5.35(±1.26) | 0.43 (±0.07) | 0.47(±0.09) |
| 51~60s | 24.39(±2.50) | 22.93(±2.71) | 13.86(±2.47) | 11.70(±2.01) | 5.19(±1.10) | 5.61(±0.98) | 0.42 (±0.08) | 0.48(±0.06) |
| 61~70s | 25.03(±2.90) | 23.23(±3.20) | 15.33(±3.10) | 12.76(±2.40) | 4.83(±1.04) | 5.25(±1.36) | 0.38 (±0.07) | 0.45(±0.09) |
| 71~80s | 24.66(±3.32) | 23.46(±3.06) | 15.70(±2.89) | 13.63(±2.55) | 4.81(±1.06) | 4.92(±1.23) | 0.36 (±0.07) | 0.41(±0.08) |
| 81~90s | 24.00(±2.96) | 23.05(±2.98) | 15.63(±2.63) | 13.75(±2.15) | 4.18(±1.06) | 4.62(±1.45) | 0.34 (±0.07) | 0.39(±0.08) |
| >90s | 24.13(±2.15) | 22.96(±2.33) | 16.15(±2.28) | 14.15(±2.46) | 3.99(±1.04) | 4.40(±1.62) | 0.33 (±0.07) | 0.37(±0.11) |
| N = 600 | 24.36 | 22.971 | 15.2853 | 13.04 | 5.18 | 4.97 | 0.37 (±0.08) | 0.433(0.91) |
| P-value | **0.000** | **0.000** | **0.000** | **0.000** | **0.000** | **0.000** | **0.000** | **0.000** |

$P < 0.05$ is considered statistically significant.
OD: Outer diameter, ID: Inner diameter, LT: Lesser trochanter, CTI: Cortical thickness index.

several studies comparing morphology between populations. Mahaisavaria et al. compared the morphology of the proximal femur between Siamese and Caucasian populations [18]. Hoaglund and Low observed different proximal geometries between the Hong Kong Chinese and Western populations [24]. Although this research was not a comparison study, the large data can be a representative of proximal femoral geometry in East Asia because Korean and Korean Chinese populations were included in this study.

Bone strength depends on both bone quantity and quality. Bone quantity can be estimated in clinical settings through BMD measurements but not quality. Bone quality encompasses the structural and material properties of bone [25]. In the clinical field, BMD and geometry could achieve bone strength using DXA and cortical thickness using radiography [20]. CT is a possible modality for analyzing the cortex in a three-dimensional manner. In several studies, a model-based approach for measuring the cortical bone thickness and density from clinical CT images has been proposed [26–28]. Our study showed average measurements of proximal femoral size as a thickness of the cortex at two different levels and proved the average changes of these indexes according to age. In addition, it has been shown that BMD in trochanteric fracture was a better predictor than that in femoral neck fracture [29]. Through comparison, the data of the trochanteric fracture and femoral neck fracture groups showed that TBS-related data were fewer in the femoral neck fracture group, i.e., a lower score of TBS could increase the risk of femoral neck fracture. A low BMD is reportedly closely related to hip fracture. The risk of hip fracture increases by 2.6-fold for each standard deviation decrease in BMD [30]. In this research, through the age- and sex-adjusted comparison between the non-fracture and hip fracture groups, the fracture group showed a lower T-score of the hip, a narrower outer diameter of the proximal femur at LT level, and thinner cortex at LT level. In the geometry field, the larger size of the outer diameter and the cortical thickness of LT could reduce the occurrence of hip fracture.

Trochanteric and femoral neck fractures are both common types of hip fractures. The average age of patients with trochanteric fracture was 5 years older than that of patients with femoral neck fracture [31]. The BMD reduction rate of the femoral neck is 0.64% before the age of 65 years and 0.36% after this age [32]. Greenspan et al. reported that BMD of the femoral neck is the most sensitive index for predicting hip fracture. The trochanteric BMD was 13% lower in women and 11% lower in men for patients with trochanteric fracture than in those with femoral neck fracture [33]. Li et al. indicated that there was no significant difference between the BMD of the trochanteric and femoral neck fracture groups [34]. In this study, through comparison between two groups of trochanteric and femoral neck fractures, the lower T-scores of the hip and spine are more susceptible to trochanteric fracture. In the aspect of CT measurements, the inner diameter of the proximal femur at 3 cm below the level LT and the CTI at the LT level were different.

The geometry of the hip also changes with aging. Patients aged ≥50 years may experience thinning of the cortex of the proximal femur and an increase in the medullary cavity [35] as well as changes in structural strength in the hip. Cortical thickness and density are critical components in determining the strength of bony structures [36,37]. Michelotti and Clark indicated that the differences between the fracture group and the controls were a thinner femoral cortex (measured at a point one head radius below the LT level), larger femoral head, and larger femoral neck diameter in the fracture group [38]. In this study, the CTI of inner diameter at 3 cm below the LT level and the cortical thickness of the LT level were significant in the two groups. The larger inner diameter of the proximal femur at 3 cm below the LT level and smaller CTI at LT level were more liable to occur in the trochanteric fracture group. Kanis et al. found that the age-adjusted risk for any type of fracture increased significantly with lower BMD [30]. The risk ratio per unit of higher BMD was 0.98 (95% confidence interval, 0.97–

0.99) for any fracture and 0.93 (95% confidence interval, 0.91–0.94) for hip fracture. The risk ratio per unit change in BMD was very similar in men and women [39]. In this study, the lower index of BMD was significantly associated with femoral neck fracture.

There are some limitations to this study. First, it was a retrospective study with a relatively small number of patients. In addition, some of the patients included in the study were missing some data, so there was a change in the number when statistical analysis was performed.

Second the DXA of patients and pelvic CT were not all completed on the same day in the clinic. The duration between the examinations was accomplished within 7 days, which means results can be affected by this. The best results can be obtained on the same day of DXA and CT examinations. Third, the authors did not concern the bony symmetry. The femur's bilateral asymmetry was reported in the previous study [40]. Fourth, the patients included were heterogenous. A more proper and accurate geometry of the proximal femur should be obtained from the homogenous group, such as the whole non-fracture group. Based on a larger amount of data, it can be more helpful to analyze the correlation between cortical bone thickness and fracture according to gender and age.

## Conclusions

Through the collection of 600 patients with hip geometry data on pelvic CT, we can represent the average data of the proximal femur in South Korea, which can reflect the data of East Asian people. CT data from the LT level and 3 cm below it was different and decreased with respect to age.

The T-scores of the hip in the fracture group were lower than that of the non-fracture group. The outer diameter of the proximal femur at the LT level was narrower, whereas the cortical thickness was thinner in the fracture group. Lower T-scores, narrower outer diameter of the proximal femur and thinner cortical thickness at the LT level can be risk factors for hip fracture.

Among the hip fracture groups, BMI, T-scores of the hip and spine, inner diameter at 3 cm below the LT level, CTI 3 cm below it, and TBS were statistically significant.

CT-based data can be comparable to DXA-based data in the aspect of fracture risk prediction. This information can also support the concept of 3D printing of implants such as intramedullary nail or stem for arthroplasty.

## Author Contributions

**Data curation:** Xiao Ma, Jungyoun Kim, Hyunjin Park, Younghyun Yoon.

**Formal analysis:** Sewon Lee, Hyunjin Park, Younghyun Yoon.

**Methodology:** Ik Yang, Hyunjin Park.

**Software:** Sewon Lee.

**Supervision:** Jungyoun Kim.

**Validation:** Younghyun Yoon.

**Writing – original draft:** Jihyo Hwang.

**Writing – review & editing:** Jihyo Hwang.

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
