## [Decision Letter · Decision Letter 0]

1 Sep 2023

PONE-D-23-00165Evaluation of cortical thickness and cortical thickness index in the proximal femur

- CT, dual energy absorptiometry (DXA), trabecular bone score (TBS) -PLOS ONE

Dear Dr. Hwang,

Thank you for submitting your manuscript to PLOS ONE. After careful consideration, we feel that it has merit but does not fully meet PLOS ONE’s publication criteria as it currently stands. Therefore, we invite you to submit a revised version of the manuscript that addresses the points raised during the review process.

We look forward to receiving your revised manuscript.

Kind regards,

Dalia Galal Mahran

Academic Editor

PLOS ONE

Journal Requirements:

"No"

"NO authors have competing interests"

5. Please upload a new copy of Figure 1 as the detail is not clear. Please follow the link for more information: "" ext-link-type="uri" xlink:type="simple">https://blogs.plos.org/plos/2019/06/looking-good-tips-for-creating-your-plos-figures-graphics/""
"" ext-link-type="uri" xlink:type="simple">https://blogs.plos.org/plos/2019/06/looking-good-tips-for-creating-your-plos-figures-graphics/""

6. Please include a caption for figure 1. 

Additional Editor Comments:

Dear authors

Thank you for the done work. Ihave comments for needed major revisins to be done.

The revisions as follows:

Methods:

• The study design is a cross sectional study made from patient records and not from previous cross sectional study. Correct the study design

• It’s better to write the methods section into subtitles, it was written in non sectional unclear different ideas like “ study population with inclusion and exclusion criteria and study sitting, data collection, statistical analysis, …………………

• Stsastical analysis: is a section under methods sections. The details of analysis are deficient

Results: All titles of the tables are deficient and some are wrong as “ Table 3:. The statistical significance in two groups “

• The tests of significance were not mentioned as footnotes

• Correct the significance from 0.000 to 0.001

• No correlation test between measurements was done

• Multivariate analysis was not done

- Discussion:

• No strengths and limitations were included

• No recommendations for further studies

Reviewers' comments:

Reviewer's Responses to Questions

**Comments to the Author**

1. Is the manuscript technically sound, and do the data support the conclusions?

Reviewer #1: Partly

Reviewer #2: Yes

2. Has the statistical analysis been performed appropriately and rigorously? 

Reviewer #1: No

Reviewer #2: No

3. Have the authors made all data underlying the findings in their manuscript fully available?

Reviewer #1: Yes

Reviewer #2: Yes

4. Is the manuscript presented in an intelligible fashion and written in standard English?

Reviewer #1: Yes

Reviewer #2: Yes

5. Review Comments to the Author

Reviewer #1: The manuscript provides the cortical thickness and and cortical thickness index from CT scan, T-score and TBS data based on DXA in total of 600 patients from South Korea. These data source is valuable in 3D printing. The findings on correlation between fracture and bone health are consistent with the literature.

The statistical analysis section mentioned all parameters were compared by Student's t-test. It works for the comparing two groups (Table 2,3,4). However, the Table 5 and table 6 listed 6 age groups, please specify which statistical tests were used for calculating p-value.

Table 1 present the average data of proximal femoral geometry from CT. Authors need to specify what are the numbers follow the average, are they range of the mean i.e. (min ~max)? Also, considering the different body size between male and female, it is necessary to provide the average data by sex.

Reviewer #2: The work presents a retrospective analysis of the proximal femur that can bring important clinical information, however the analysis of some parameters can improve the work.

The purpose of the mentioned study should be changed in order to encompass all the analysis carried out. It seems to me that changing to: Characterizing the proximal femur of patients without alterations, with fractures...in different parameters...would be more appropriate.

How did the sample of 200+200+200 arrive, was it by chance? Explain the methodology in detail.

Show pictures of examples of trochanteric fracture and femoral neck fracture.

Tables must be self-explanatory and in the title mention the type of sample used

The data with means must be accompanied by the SD

Statistically, age and gender should also be studied in the different parameters analyzed, using only the sample of 200 normal patients as a reference. Therefore, a different and more complete statistical analysis is recommended.

6. PLOS authors have the option to publish the peer review history of their article (what does this mean?). If published, this will include your full peer review and any attached files.

Reviewer #1: No

Reviewer #2: No

---

## [Author Response · Author response to Decision Letter 0]

10 May 2024

Respond to academic editor 

1. Please ensure that your manuscript meets PLOS ONE's style requirements, including those for file naming

= We changed Scipts style considering on PLOS manual style templit 

: Followed Title , Author , Affiliation formating guideline etc. 

2. At this time, please address the following queries

“The authors received no specific funding for this work”

= we included this comment on cover letter

"NO authors have competing interests"

= “The authors have declared that no competing interests exist. “ was included on cover letter

4. Data availability

= We uploaded our patient data on figshare and it’s DOI is 10.6084/m9.figshare.24755277

5. Please upload a new copy of Figure 1 as the detail is not clear.

= Figure 1 ‘s resolution has been changed and upload a new copy 

6. Please include a caption for figure 1.

= Added caption for figure 1 

7. Please include captions for your Supporting Information files at the end of your manuscript, and update any in-text citations to match accordingly.

= We changed all caption for supporting information and uploaded end of our manuscripts. it also match accordingly 

Additional respond to academic editor 

Methods:

• The study design is a cross sectional study made from patient records and not from previous cross sectional study. Correct the study design

= We modified the study design to fit 

• It’s better to write the methods section into subtitles, it was written in non sectional unclear different ideas like “ study population with inclusion and exclusion criteria and study sitting, data collection, statistical analysis, …………………

= The method section was modified with subtitles and statistical analysis was moved to the method section.

• Stsastical analysis: is a section under methods sections. The details of analysis are deficient

= statistical analysis was moved to the method section. and supplemented the details.

Results: All tis “

• All titles of the tables are deficient and some are wrong as “ Table 3:. The statistical significance in two groups “

= all table’s title was changed and made them fit 

• The tests of significance were not mentioned as footnotes

= The statement related to the table was written on footnote.

• Correct the significance from 0.000 to 0.001

= We fixed it 

• No correlation test between measurements was done

• Multivariante analysis was not done

= It was difficult to conduct correlation tests and multivariant analysis studies because there were many missing information due to insufficient samples. However, this study is a meaningful paper in analyzing the correlation between cortical bone thickness and fractures in Far East Asia. It is unfortunate that the above research was omitted, but it was difficult due to the limitations of sample data.

- Discussion:

• No strengths and limitations were included

• No recommendations for further studies

= We added limation and recommendation for futher studies

Respond to reviewer

Reviewer #1: The manuscript provides the cortical thickness and and cortical thickness index from CT scan, T-score and TBS data based on DXA in total of 600 patients from South Korea. These data source is valuable in 3D printing. The findings on correlation between fracture and bone health are consistent with the literature.

The statistical analysis section mentioned all parameters were compared by Student's t-test. It works for the comparing two groups (Table 2,3,4). However, the Table 5 and table 6 listed 6 age groups, please specify which statistical tests were used for calculating p-value. 

= Not all were done with study t-test. The statistics of tables 5 and 6 you mentioned were modified to use ANOVA because there was a mistake when writing it.

Table 1 present the average data of proximal femoral geometry from CT. Authors need to specify what are the numbers follow the average, are they range of the mean i.e. (min ~max)? 

= I wrote the part you told me ( min ~ max ) on footnote.

Also, considering the different body size between male and female, it is necessary to provide the average data by sex. Fourth, the patients included were heterogenous. A more proper and accurate geometry of the proximal femur should be obtained from the homogenous group, such as the whole non-fracture group. 

= As a retrospective study conducted with a limited sample, it was difficult to perform the classification according to the gender you mentioned. If research based on other samples is conducted in the future, it is believed that academic achievement will be achieved through meta-analysis.

Reviewer #2: The work presents a retrospective analysis of the proximal femur that can bring important clinical information, however the analysis of some parameters can improve the work.

The purpose of the mentioned study should be changed in order to encompass all the analysis carried out. It seems to me that changing to: Characterizing the proximal femur of patients without alterations, with fractures...in different parameters...would be more appropriate.

= What we wanted to show was to study the correlation between fractures and anatomical features, including femoral proximal cortical bone thickness. This required a comparison between fracture and non-fracture patients. There were relatively many data on fracture patients in hospitals, so it was possible to measure and compare the opposite side, but in the case of non-fracture patients, it was difficult to perform CT and other tests for the necessary data

The non-facture group was formed by collecting data from bone necrosis patients of the femoral head or suspected fracture patients who came to the outpatient clinic, and due to the nature of this medical institution, sufficient data was not accumulated, so it is regrettable that gender comparisons were not made. However, the purpose of the study was to correlate the fracture with the anatomical values of the proximal femur.

How did the sample of 200+200+200 arrive, was it by chance? Explain the methodology in detail.

= We changed the method section to However, among the patients who visited our hospital without fractures, it was very difficult to find patients who performed all the necessary tests when we studied. Therefore, among patients without fractures, 200 patients who had completed the examination were collected and compared with patients with fractures.

Show pictures of examples of trochanteric fracture and femoral neck fracture.

= We attached the femur neck fracture as figure 2 and the trochanetic fracture as figure 3, and the original figures 2,3,4,5,6,7 were revised to 4,5,6,7,8,9 respectively.

Tables must be self-explanatory and in the title mention the type of sample used

= We have added a new detailed description of the table to the footnote at the bottom of the table

Statistically, age and gender should also be studied in the different parameters analyzed, using only the sample of 200 normal patients as a reference. Therefore, a different and more complete statistical analysis is recommended.

= It was difficult to conduct different and more complete statistical analysis studies because there were many missing information due to insufficient samples. However, this study is a meaningful paper in analyzing the correlation between cortical bone thickness and fractures in Far East Asia. It is unfortunate that the above research was omitted, but it was difficult due to the limitations of sample data.

---

## [Decision Letter · Decision Letter 1]

12 Jun 2024

PONE-D-23-00165R1Evaluation of cortical thickness and cortical thickness index in the proximal femur

- CT, dual energy absorptiometry (DXA), trabecular bone score (TBS) -PLOS ONE

Dear Dr. Hwang,

Thank you for submitting your manuscript to PLOS ONE. After careful consideration, we feel that it has merit but does not fully meet PLOS ONE’s publication criteria as it currently stands. Therefore, we invite you to submit a revised version of the manuscript that addresses the points raised during the review process.

We look forward to receiving your revised manuscript.

Kind regards,

Alessandra Aldieri

Academic Editor

PLOS ONE

Journal Requirements:

Additional Editor Comments:

There are still a couple of changes to be implemented prior to publication

Reviewers' comments:

Reviewer's Responses to Questions

**Comments to the Author**

1. If the authors have adequately addressed your comments raised in a previous round of review and you feel that this manuscript is now acceptable for publication, you may indicate that here to bypass the “Comments to the Author” section, enter your conflict of interest statement in the “Confidential to Editor” section, and submit your "Accept" recommendation.

Reviewer #1: All comments have been addressed

2. Is the manuscript technically sound, and do the data support the conclusions?

Reviewer #1: Yes

3. Has the statistical analysis been performed appropriately and rigorously? 

Reviewer #1: Yes

4. Have the authors made all data underlying the findings in their manuscript fully available?

Reviewer #1: Yes

5. Is the manuscript presented in an intelligible fashion and written in standard English?

Reviewer #1: Yes

6. Review Comments to the Author

Reviewer #1: In the Data collection section, author stated “propensity score matching was performed to compare the data of the two groups” (i.e. hip fracture group and non-fracture group). Suggest authors specifying this selection method in the table 2 footnote and describing what parameters were included in the propensity score calculation.

For Table 6, authors need to specify what are the numbers after the +/- sign. The presentation of N=600 row is not consistent across the 8 columns. The last two columns showed mean (+/-) but not the other 6 columns. Authors shall fix the inconsistency.

7. PLOS authors have the option to publish the peer review history of their article (what does this mean?). If published, this will include your full peer review and any attached files.

Reviewer #1: No

---

## [Author Response · Author response to Decision Letter 1]

20 Jul 2024

Respond to reviewer

Reviewer #1: In the Data collection section, author stated “propensity score matching was performed to compare the data of the two groups” (i.e. hip fracture group and non-fracture group). Suggest authors specifying this selection method in the table 2 footnote and describing what parameters were included in the propensity score calculation.

Thank you for your good question, we included only age and sex factors for the propensity score matching. We added comment about this in the manuscript.

For Table 6, authors need to specify what are the numbers after the +/- sign. 

+/- sigh is standard deviation, we included explanation about this

The presentation of N=600 row is not consistent across the 8 columns.

Thank you for your sharp point, we corrected this at the table 6

 The last two columns showed mean (+/-) but not the other 6 columns. Authors shall fix the inconsistency

Thank you for your detail comments, we added standard deviation to the other 6 columns

---

## [Editor Report · Decision Letter 2]

25 Jul 2024

Evaluation of cortical thickness and cortical thickness index in the proximal femur

- CT, dual energy absorptiometry (DXA), trabecular bone score (TBS) -

PONE-D-23-00165R2

Dear Dr. Hwang,

We’re pleased to inform you that your manuscript has been judged scientifically suitable for publication and will be formally accepted for publication once it meets all outstanding technical requirements.

Kind regards,

Alessandra Aldieri

Academic Editor

PLOS ONE
---

## [Editor Report · Acceptance letter]

13 Nov 2024

PONE-D-23-00165R2 

PLOS ONE

Dear Dr. Hwang, 

I'm pleased to inform you that your manuscript has been deemed suitable for publication in PLOS ONE. Congratulations! Your manuscript is now being handed over to our production team.

Kind regards, 

on behalf of

Dr. Alessandra Aldieri 

Academic Editor

PLOS ONE